# Intravitreal 0.19 mg Fluocinolone Acetonide Implant in Non-Infectious Uveitis

**DOI:** 10.3390/jcm10173966

**Published:** 2021-09-01

**Authors:** Muaas Hikal, Nil Celik, Gerd Uwe Auffarth, Lucy Joanne Kessler, Christian Steffen Mayer, Ramin Khoramnia

**Affiliations:** Department of Ophthalmology, University Hospital of Heidelberg, 69120 Heidelberg, Germany; muaas.hikal@med.uni-heidelberg.de (M.H.); nil.celik@gmx.de (N.C.); gerd.auffarth@med.uni-heidelberg.de (G.U.A.); LucyJoanne.Kessler@med.uni-heidelberg.de (L.J.K.); Christian.Mayer@med.uni-heidelberg.de (C.S.M.)

**Keywords:** non-infectious uveitis, uveitic macular edema, 0.19-mg fluocinolone acetonide implant

## Abstract

The efficacy of the 0.19-mg fluocinolone acetonide (FAc) intravitreal implant (ILUVIEN) in the treatment of non-infectious uveitic macular edema (UME) was assessed on twenty-six patients (34 eyes) with non-infectious UME between 2013 and 2020, in a mean follow-up of 18 ± 19.3 (mean ± SD) months. Macular edema was resolved in 24 (70.6%) cases. Five of these eyes had a relapse after 23.2 ± 14 months. Three FAc reinjections were performed and a drying of the macula was observed. After FAc implantation, 24 eyes (70.6%) were completely dry; central retinal thickness (CRT) decreased in 6 eyes (17.6%), but residual intraretinal fluid was still evident. In 20 eyes (58.5%), visual acuity (VA) improved (from +1 to +5 lines) and remained stable in 9 eyes (26.5%). Thirty eyes (88.2%) were pseudophakic at baseline and four were phakic. Three of these eyes had a cataract prior to therapy and the other developed a cataract 2.5 years after the FAc implant was administered. There was an overall increase in intraocular pressure (IOP; +4.4 ± 3.7 mmHg) and eye drops were required in three eyes. The FAc implant led to long-term improvements in mean CRT and VA, and that the side-effect profile was manageable in a clinical setting in patients with non-infectious UME.

## 1. Introduction

Cystoid uveitic macular edema is caused by the breakdown of the blood-retina barrier due to inflammation and is a main factor for the loss of vision in patients with non-infectious uveitis affecting the posterior segment of the eye [1,2]. The treatment of uveitis with corticosteroids (both systemically and/or locally) have been shown to be effective in reducing the extent of inflammation and macular edema [3,4,5]. Topical therapy has been shown to be effective in the treatment of anterior uveitis, but less so for other forms of uveitis such as intermediate uveitis, posterior uveitis and panuveitis [3]. In such cases, topical therapy is combined with systemic therapy, especially where there is systemic involvement, and often associated with multiple side effects and a heavy treatment burden for the patient [3,6,7].

Sustained-release intravitreal corticosteroid implants have gained growing interest clinically as they provide long-term therapy and help to reduce the number of treatments and treatment visits [8]. The 0.7 mg dexamethasone implant (Ozurdex) lasts up to 6 months and has been successfully used in the treatment of uveitis; however, it does require repeated injections [5,9]. The 0.19 mg ILUVIEN (fluocinolone acetonide [FAc] Alimera Sciences Inc., Alpharetta, GA, USA) intravitreal implant consists of a non-biodegradable polymer with a length of 3.5 mm and a diameter of 0.37 mm and provides a daily sustained release of 0.2 µg fluocinolone acetonide for up to 3 years. The treatment of diabetic macular edema (DME) with the FAc implant was shown to be efficacious in the Fluocinolone Acetonide for Diabetic Macular Edema (FAME) studies and licensed to treat DME [10,11]. In 2019, the FAc implant was licensed for the for prevention of relapse in recurrent non-infectious uveitis affecting the posterior segment of the eye [12].

The FAc implant has been shown to be effective in reducing the number of uveitis recurrences as well as prolonging the time between treatment and the overall number of treatments required over 3 years [13]. Our retrospective observations study was designed to evaluate the effectiveness and safety of the use of the FAc implant in non-infectious uveitis affecting the posterior segment of the eye in our practice in a rather long follow-up in a real-world setting.

## 2. Methods

We conducted a retrospective chart review involving 34 eyes (26 patients) with non-infectious uveitic macular edema (UME) who were treated in the Department of Ophthalmology at the University of Heidelberg, Germany. Ethical approval was obtained from the local ethical committee of the University of Heidelberg (S-644/2020). The study was registered in the German clinical trials register (DRKS00024399). To exclude infectious uveitis, patients were tested for toxoplasmosis, syphilis, tuberculosis, herpes, cytomegalovirus, Lyme disease, or HIV, depending on the morphology of the uveitis. Patients had a history of uveitic macular edema (for up to 13 years) and were being treated with systemic and/or local corticosteroids—topical, periocular or intravitreal—prior to therapy with the FAc implant. Patients with non-infectious uveitic macular edema have their intravitreal FAc implantations reimbursed and the drug has a label for this disease. Thus, the FAc implantiotion was therefore preferred to simple topical therapy, orbital injections, or subconjunctival injections. Patients were treated with the FAc implant between 2013 and 2020 (note: it was used off-label prior to its approval in Europe in 2019).

Study parameters included the measurement of central retinal thickness (CRT), which was measured using spectral domain optical coherence tomography (SD-OCT; Heidelberg Engineering); corrected distance visual acuity (VA); uveitic activity; and, signs of intraocular inflammation in different segments of the eye according to the SUN Working Group’s grading of ocular inflammation in uveitis [14]. Other parameters evaluated included the time to relapse, the change of intraocular pressure (IOP), measured using Goldmann applanation tonometry, and cataract development. Clinical examinations were performed over a mean period of 1 to 60 months (18 ± 19.3 [mean ± SD] months).

Data are reported as mean ± standard deviation (SD) unless otherwise stated.

## 3. Results

The mean age at baseline was 58 ± 15.5 years (range, 26 to 87 years). Eighteen (69.2%) out of the 26 participants were females. The mean duration of UME was 6.9 ± 4.0 years (range, 1 to 15 years). Table 1 shows the details of uveitis localization and its etiology and Table 2 shows the local corticosteroid therapies given prior to injection of FAc implant. The off-label use of the 0.59 mg fluocinolone acetonide implant (Retisert; Bausch & Lomb Inc., Rochester, NY, USA) was also employed in one eye with intermediate uveitis and was given 4.5 years prior to the FAc implant. In 2003 the same patient had a subsequent fluocinolone acetonide implant, which was removed in 2008 due to protrusion of the implant.

### 3.1. Central Retinal Thickness (CRT)

#### 3.1.1. Overall Effect of FAc Therapy

A total of 37 eyes was identified including three reinjections. Three of these subsequently failed to attend follow-up appointments. One to 3 months after injection of the FAc implant, a decrease in CRT was observed in 27 eyes.

Twenty-four (70.6%) out of the 34 injected eyes showed a complete drying of the macula and in 6 eyes (17.6%) a decrease in macular edema was recorded but without complete drying of the macular. The other four eyes (11.8%) did not experience an improvement in CRT (Figure 1). Figure 2 shows the time-course for the changes in CRT following therapy with the FAc implant.

The OCT images in Figure 3 show an example of macular edema that regressed after injection, the macula remaining dry at 60 months follow up. Figure 4 shows a severe macular edema that was completely dry after treatment with the FAc implant. Supplementary injections of a dexamethasone intravitreal implant (x1 injection), triamcinolone parabulbar (x1) and subconjuntival triamcinolone (x6) were given in the first year. After the macula was completely dry, no further therapy was necessary, so that the reinjection of FAc did not occur until 32 months after initial injection. After the second injection, the macula remained dry.

#### 3.1.2. Relapse and Retreatment with a FAc Implant

During treatment, five eyes (14.7%) with a dry macula experienced a relapse in UME, of which three eyes were re-injected. The first eye was retreated after 42 months and the patient subsequently did not attend their next appointment. Weber et al. mentioned this patient as case 4, whose additional data were further processed in this work [12].

The second patient eye, which required a second implant, was retreated after 32 months. After the first FAc implant was injected, CRT decreased from 930 µm to 485 µm. Supplemental periocular injection of triamcinolone was given to dry the macula and this approach was repeated with supplementary subconjunctival triamcinolone injections to maintain a dry macula. Following the second FAc implant, the macula remained dry with no supplemental therapy required. This patient was mentioned as case 3 in Weber et al. [12].

The third eye experienced a relapse after 18 months and this was controlled with supplemental periocular injection of triamcinolone and a dexamethasone implant.

The fourth eye had a dry macula for 18 months. During this time no additional corticosteroid injections (of triamcinolone or the dexamethasone implant) were given due to a previously defined IOP response to corticosteroid. A second FAc implant was given at 18 months to control the relapse.

In the fifth eye, a relapse occurred after a YAG capsulotomy was performed at month 6 and the macula was dried effectively with a single periocular injection of triamcinolone. The overall mean duration of recurrence was 23.2 ± 14 months.

#### 3.1.3. Therapy Failures

One patient showed up 3 months after FAc implantation but refused to have a SD-OCT. Using fluorescein angiography [FA] we detected a decrease macula edema without being able to quantify the edema as an OCT was refused. One week later, the patient presented with a retinal detachment. Furthermore, two eyes, which showed no decrease in CRT, had no improvement to repeated injections of the dexamethasone implant, but improvements were evident following injection of the FAc implant.

Another patient had an increase in macula edema following intravitreal injection of the FAc implant, even though they had temporary macula drying following treatment with an intravitreal dexamethasone implant.

### 3.2. Visual Acuity (VA)

#### 3.2.1. Overall Effect of FAc Therapy

Figure 5 shows the maximum change in VA after the injection of the FAc implant and that after the injection of the FAc implant, 20 eyes (58.8%) experienced an improvement (2 ± 1.2 lines; range, +1 to +5 lines) in VA, that VA remained stable in 9 eyes (26.5%) and worsened in 5 eyes (14.7%) (range, 1 to 2 lines). We here present the maximum visual acuity gain in comparison to baseline.

#### 3.2.2. Stable VA after FAc Therapy (n = 9)

The reasons for the stable VA in nine of the eyes can be explained. Two of the eyes had a marked improvement in VA (to 0.8 decimal) prior to administration of the FAc implant and two further eyes had signs of permanent retinal damage (due to IS/OS layer disruption and a prior retinal vein occlusion). Four eyes had ME and no reduction in CRT. The last eye showed a complete drying of the macula with improved visual acuity, compared to previous visits, and was injected with a FAc implant due to the risk of relapse.

#### 3.2.3. Worsening of VA after FAc Therapy (n = 5)

Two out of the five eyes that deteriorated showed a complete drying of the macula, but one of them had simultaneous IS/OS damage and the other had an epiretinal gliosis. A further two eyes showed no reduction in CRT. The fifth eye lost two lines after FAc implant despite a minor reduction in CRT while showing an epiretinal gliosis and a progressive subcapsular posterior cataract.

Over time, these five eyes, which had showed an initial improvement, started to partly worsen for several reasons. After 3.5 years, one eye showed an increase in CRT and required retreatment but the patient failed to attend the clinic after reinjection. One eye had a total retinal detachment and could only detect hand motion. After gaining five lines post-FAc implantation, one eye showed a notable relapse after a YAG capsulotomy. A further two eyes had fluctuating VA and CRT.

### 3.3. Uveitis Activity

During the follow-up examination the signs of inflammation were evaluated according to the grading of ocular inflammation in uveitis by the SUN Working Group [14]. Figure 6 shows the uveitis activity before and 1–3 months after the injection of the FAc implant and shows a notable decrease in inflammation in both the anterior and posterior segments of the eye.

### 3.4. Systemic Therapy

As well as local therapy, 14 (53.8%) of 26 patients required systemic therapy, which during follow-up was reduced or discontinued according to the underlying condition. Table 3 provides an overview of the changes in treatment pre- and post-FAc implantation.

### 3.5. Adverse Events

#### 3.5.1. Intraocular Pressure

Figure 7 plots the course of IOP following FAc implantation. Following therapy, there was a mean increase in IOP of 4.4 ± 3.7 mmHg (range, 1 to 14) in 23 eyes.

At baseline, 5 of the 34 eyes were being treated with one IOP-lowering drop and supplemental eye drops were required in three of these five eyes postinjection of the FAc implant (Table 4). These eyes were effectively managed with IOP after FAc therapy. Indeed, the first patient was receiving brinzolamide as IOP lowering drops in both eyes. At month 15, the patient presented to the clinic and, despite a time difference of 15 months between FAc implantation in left and right eyes, experienced elevations in IOP in both eyes (from 8 to 18 mmHg in the left eye and from 12 to 26 mmHg in the right eye). Both eyes were managed with additional brimonidine eye drops and then latanoprost eye drops were given four months later.

The second patient had also been treated bilaterally in the past. The left eye had undergone several anti-glaucomatous procedures (i.e., trabeculectomy, iridectomy) before the injection of the FAc implant. At the time of the injection, neither the right nor the left eye was treated with IOP lowering eye drops. The pressure in the right eye, however, increased from 16 to 23 mmHg and an additional IOP drop of dorzolamide was required. The intraocular pressure of the left eye did not increase to values that needed therapy.

#### 3.5.2. Cataract Formation

At baseline, 4 of 34 injected eyes were phakic and prior to FAc implantation had signs of cataract development. The fourth eye underwent cataract removal surgery 2.5 years after the injection of the FAc implant.

#### 3.5.3. Retinal Detachment

Three months after FAc injection, a patient appeared for the first follow-up with an increased VA and reduced ME, detected using FA, but refused a SD-OCT. One week later the patient presented with a total retinal detachment. Intraoperatively, no retinal tear could be found as the cause of a possible rhegmatogenous retinal detachment. A significant opacity and flat white subretinal plaques were detected and the FAc implant was removed as part of the vitrectomy procedure.

#### 3.5.4. Hypotonia

Two patients showed clear hypotonia with choroidal folds immediately after FAc implantation. The follow-up showed a stable and increased IOP with a notable decrease in the choroidal folds. It should be mentioned that one of these eyes had multiple surgeries in the same eye prior to the FAc implant.

## 4. Discussion

In our work, local therapy with 0.19 mg FAc implant led to a decrease in UME over time in most patients. Prior to injection, improvements could only be achieved with multiple injections or additional systemic therapy. The FAc implant also helped to control the uveitic activity and adverse events (i.e., IOP elevation and cataract formation) were manageable.

The normal approach to managing UME has been the use of local and systemic glucocorticoids. In order to avoid the systemic side effects, there is an increasing tendency to treat the disease locally. Since UME occurs more frequently with intermediate or posterior uveitis, local eyedrops have only a limited effect and the focus of treatment is on local intravitreal injections [3].

Periocular depot corticosteroid injections in patients with ocular inflammatory disorders were effective in treating active intraocular inflammation and in improving reduced VA attributed to ME. In this multicenter retrospective cohort study, 914 patients (1192 eyes) received ≥1 periocular injection during follow-up. The main outcomes were the control of inflammation, the improvement of visual acuity (VA) and the prevention of VA loss attributed to macular edema (ME). However, within 12 months cataract surgery occurred in 13.8% of eyes at risk (95% CI, 11.1–17.2) and glaucoma surgery occurred in 2.4% (95% CI, 1.4–3.9) of eyes [4].

The authors of the Huron study concluded that a single dexamethasone intravitreal injection had a significant effect on visual acuity and intraocular inflammation that lasted for up to 6 months in patients with non-infectious intermediate or posterior uveitis [5]. This study was designed to last 26-weeks and eyes with non-infectious intermediate or posterior uveitis were randomized to a single treatment with a 0.7-mg DEX implant (n = 77), 0.35-mg DEX implant (n = 76), or sham procedure (n = 76). The primary outcome measure was based on the amount of vitreous haze that obscured visualization and the proportion of patients with a vitreous haze score of 0 at week 8. The proportion of eyes with a vitreous haze score of 0 at week 8 was 47% with the 0.7-mg DEX implant, 36% with the 0.35-mg DEX implant, and 12% with the sham (*p* < 0.001), which persisted through week 26. The group with dexamethasone implant showed a gain of 15 or more letters from baseline best-corrected visual acuity which was significantly more than in the sham group throughout the study period. The percentage of eyes with intraocular pressure of 25 mmHg or more peaked at 7.1% for the 0.7-mg DEX implant, 8.7% for the 0.35-mg DEX implant, and 4.2% for the sham (*p* > 0.05 at any visit). The incidence of cataracts reported in the phakic eyes was 9 of 62 (15%) with the 0.7-mg DEX implant, 6 of 51 (12%) with the 0.35-mg DEX implant, and 4 of 55 (7%) with the sham (*p* > 0.05) [5]. The dexamethasone implant had also shown efficiency in treating persistent uveitic macula edema [15].

The treatment of non-infectious uveitis with longer-acting corticosteroids is evolving. For instance, the surgically placed 0.58 mg fluocinolone acetonide implant (Retisert) has been determined over 7 years in the Multicenter Uveitis Steroid Treatment Trial (MUST) [6]. In another trial lasting 34 weeks and involving 278 patients with recurrent non-infectious posterior uveitis, patients were randomized to receive a 0.59-mg (n = 110) or 2.1-mg (n = 168) implant surgically, the authors studied the recurrence rate, VA outcomes, the need for adjunctive therapy and safety outcomes. The 0.59-mg FA implant reduced the rate of recurrence from 51.4% in the 34 weeks preceding implantation to 6.1% postimplantation (*p* < 0.0001) versus a significant increase (from 20.3% preimplantation to 42.0% postimplantation (*p* < 0.0001)) in the recurrence rate in the fellow non-implanted eye. VA was stabilized or improved in 87% of implanted eyes and was generally associated with reductions in the area of macular hyperfluorescence in fluorescein angiography. The percentage of eyes that required systemic medications, periocular injections, and topical corticosteroids decreased from 52.9%, 63.0%, and 35.7%, respectively, preimplantation to 12.1%, 2.2%, and 16.5% postimplantation (*p* < 0.0001 in all cases). At week 34, 51.1% of implanted eyes required ocular antihypertensive drops and 5.8% underwent glaucoma filtering surgery. Lens opacity scores increased by ≥2 grades in 19.8% of phakic implanted eyes and 9.9% required cataract surgery [16].

In another study where 11 eyes of 11 participants with non-infectious intermediate uveitis, posterior uveitis, and panuveitis were followed up for 2 years, intraocular inflammation in all eyes was effectively controlled and all implanted eyes demonstrated an improvement in visual acuity. Elevations in IOP were recorded in 18% of implanted eyes and was managed by IOP-lowering drops [17].

The efficacy of 0.19 mg FAc implant in diabetic macula edema (DME) has been studied in the FAME trial, where subjects with persistent DME despite ≥1 macular laser treatment were randomized 1:2:2 to sham injection (n = 185), low-dose insert (n = 375), or high-dose insert (n = 393). In patients with DME, the 0.19 mg FAc implant was shown to provide substantial visual benefit that lasted for 3 years and the authors concluded that it provided a valuable addition to the options available for patients with DME [10].

Treatment with fluocinolone acetonide intravitreal implant was associated with significantly fewer episodes of uveitic recurrence, a significantly longer time to uveitic recurrence, greater improvement in visual acuity, a lower need for adjunctive therapy, and an acceptable safety profile [18,19].

In this work we evaluated the data of patients who were treated with the 0.19 mg FAc implant for UME and studied its effects on macular edema, central retinal thickness, uveitis activity, VA and aspects concerning its safety (i.e., IOP and cataract formation). The study showed a decrease in CRT in most cases (30 of the 34 injected eyes). In 24 eyes, the macula was completely dry and 6 eyes had drying but residual fluid persisted. In a mean follow-up period of 18 ± 19.3 months (range, 1 to 60 months), the long-term effectiveness of the drug should be emphasized. As mentioned in Table 1, all the eyes had been treated with local corticosteroids before treated with FAc implant and many patients had also required systemic therapy prior to the implant being administered. We found that eyes requiring multiple local injections prior FAc implantation were able to remain stable on FAc without further therapy for up to 23.2 ± 14 months. As Figure 2 highlights, the first follow-up showed a decrease in the central retinal thickness after 1–3 months and indicates the rapid onset of action of the implant which was maintained to month 6, although a slight increase in the mean CRT was observed and may be explained by the fact that five patients required supplemental therapy at that time. However, the majority still had low CRT values. In those eyes where supplemental therapy was required, mean CRT values were effectively managed with the combined approach to therapy. After 42 months, another increase in mean CRT can be observed and is taken as being indicative of a decrease in the effectiveness of the implant. This was expected, however, as the FAc implant has been shown to elute fluocinolone acetonide for three years [20]. Out of all the injected eyes, five showed a relapse over the time. Two eyes had an expected relapse after the expected end of effectiveness of the implant. In our cases this was observed at years 2.5 and 3, and in both cases the patient received a second FAc implant. In some cases where a relapse occurred, additional local treatment was administered, although it is worthwhile to mention that those eyes requiring additional therapy had reacted poorly to initial treatments (i.e., prior to the FAc implant) which included the dexamethasone implant and intravitreal injections of triamcinolone. Therapy with FAc implant enabled the injection intervals to be stretched, so that 2 eyes were injected with dexamethasone each at an interval of 6 and 12 months. One eye with a CRT more than 900 µm required many additional local steroids to dry the macula. After drying of the macula, the sustained dose delivered by the FAc implant was sufficient to avoid further relapse in UME and no further treatments were required. Hence, it is postulated that once the severe edema had been effectively managed with initial local treatment, the sustained daily dose of 0.2 µg FAc was sufficient to prevent the development of new macular edema and thus prevent further injections.

One of re-injected patients has not attend all follow-up visits after the injection of a second FAc implant, which was more than a year before the evaluation. Regular follow-ups with long waiting times and continuous injections at shorter intervals can be kept within limits by a long-acting therapy. Regular check-ups, injections and waiting times can seriously affect their quality of life and impact patient compliance, both for the elderly and for those in working life who must constantly take time off work.

One of the treated patients had previously received a surgical fluocinolone implant in the early 2000s and Retisert in 2013. Under treatment with that high dose, the macula was completely dry. However, it was associated with many complications, such as increased intraocular pressure requiring multiple operations and, ultimately, a dislocation of the implant, so that it had to be removed. With dexamethasone and triamcinolone, the intraocular pressure values increased greatly, which made the therapy more difficult. After injecting the 0.19 mg FAc implant, CRT decreased significantly without complete drying of the macular; however, IOP was controlled and the patient was satisfied with their therapy. In this case, the regular dose of 0.2 µg FAc per day was not enough to completely dry the macula, but it was easier to manage the previously reported side effects in this patient whilst also reducing macula edema.

Treatment with the 0.19 mg FAc implant was also associated with an increase in VA (i.e., an increase in 55.9% (n = 19) of the eyes and stability in 26.5% (n = 9) of the cases). Improvements in CRT did not necessarily correlate with improvements in VA. It should also be mentioned that many of the treated eyes have had uveitis and macula edema for many years. These patients also had partial or advanced IS/OS damage that is irreversible, which limits the potential increase in visual acuity.

The 0.19 mg FAc implant showed an acceptable and manageable risk profile. We identified no marked increases in IOP requiring surgical intervention. However, in one out of four pseudophakic eyes, cataract progression was observed after injection of the FAc implant. The retinal detachment recorded several months after the injection of the FAc implant is unlikely to be due to the injection.

In summary: our data show an overall benefit in terms of the reduction in macula edema based on recordings of CRT and in terms of VA, as well as a remarkable reduction in recurrence in a rather long follow-up of patients treated with an FAc implant in a real-world setting. Therapy with the FAc implant also offers other potential benefits, such as reductions in the need for repeated intravitreal injections and visits to the clinic. The risk profile was manageable without severe outcomes, e.g., endophthalmitis, severe hypotonia or hypertension.

## Figures and Tables

**Figure 1 jcm-10-03966-f001:**
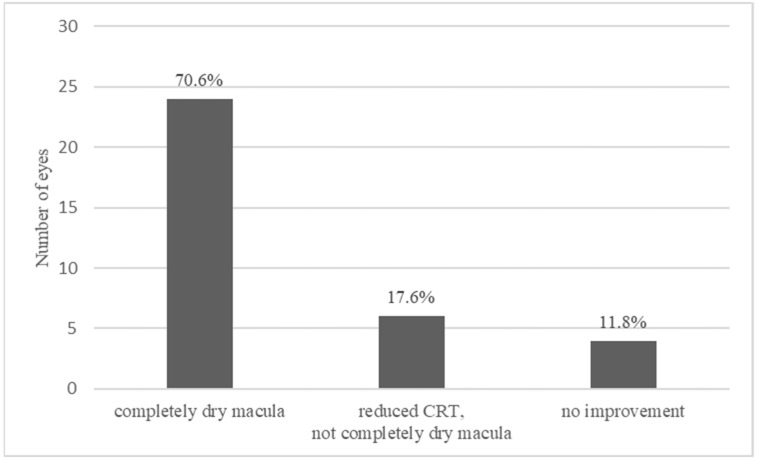
Extent of macula edema 1–3 months following the intravitreal injection of a FAc implant.

**Figure 2 jcm-10-03966-f002:**
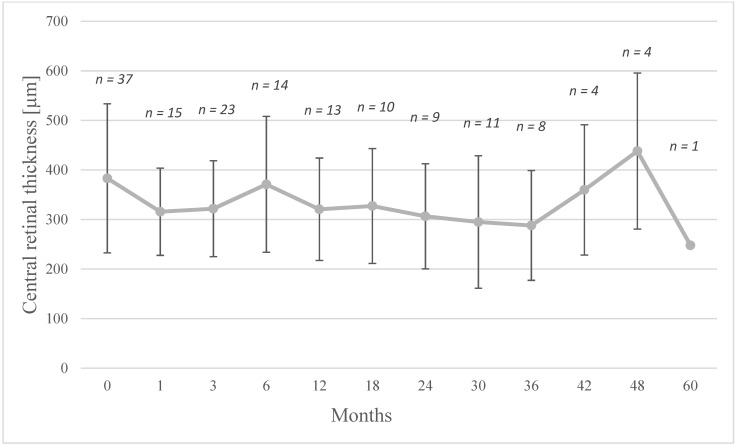
Central retinal thickness (mean ± SD) following the intravitreal injection of a FAc implant. At baseline, 37 eyes (including 3 reinjections) were included in the study; however, three failed to attend follow-up appointments after the baseline visit.

**Figure 3 jcm-10-03966-f003:**
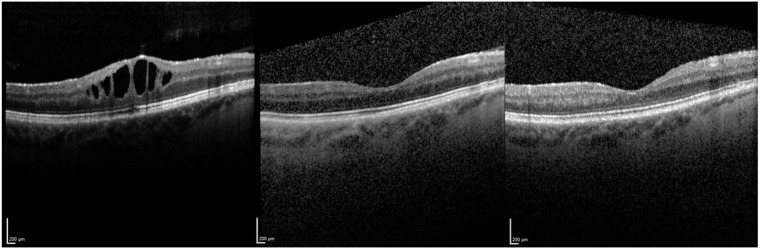
Optical coherence tomography pre FAc, 3 months post FAc and 60 months post FAc injection.

**Figure 4 jcm-10-03966-f004:**
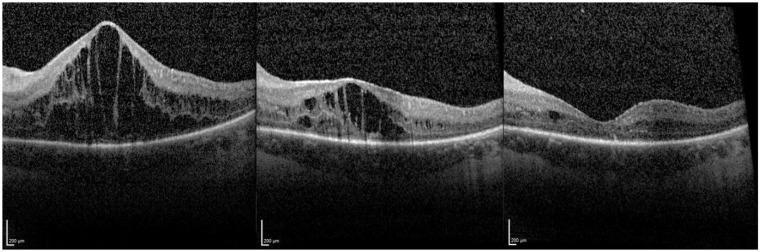
Optical coherence tomography of severe macular edema pre FAc, 3 months after FAc, and 12 months after FAc and additional single injection of Ozurdex and multiple triamcinolone injections to dry the macula.

**Figure 5 jcm-10-03966-f005:**
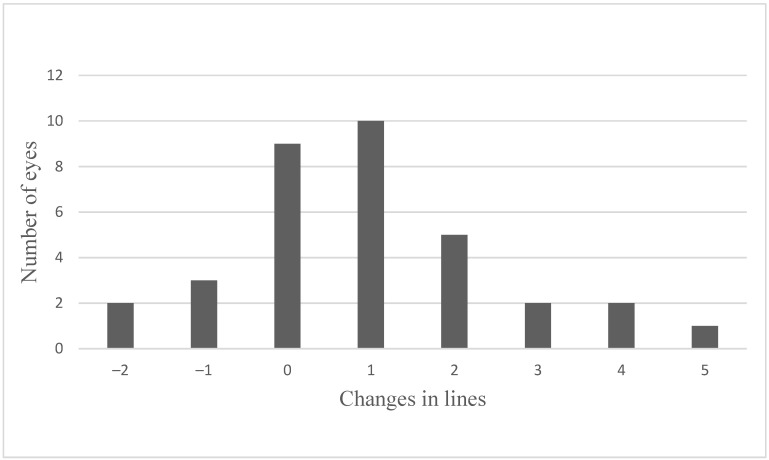
Maximal change of visual acuity in lines following the intravitreal injection of a FAc implant.

**Figure 6 jcm-10-03966-f006:**
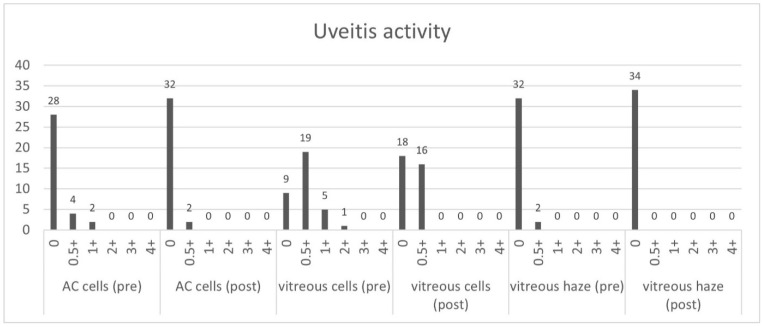
Uveitis activity pre- and post-FAc implantation.

**Figure 7 jcm-10-03966-f007:**
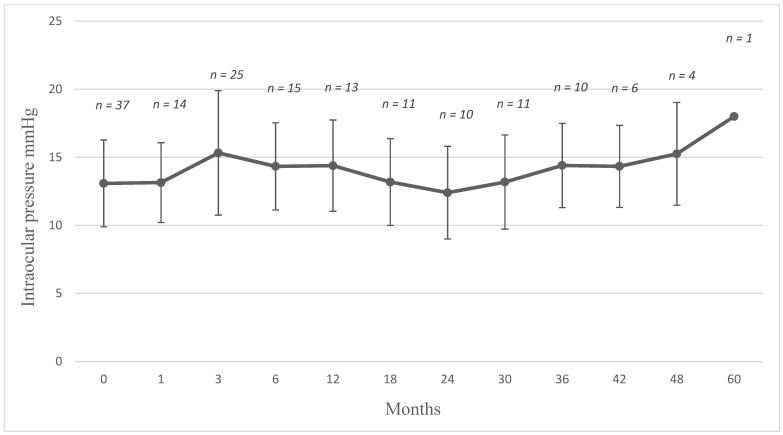
Intraocular pressure (mean ± SD) following the intravitreal injection of a FAc implant. At baseline, 37 eyes (including 3 reinjections) were included in the study; however, three failed to attend follow-up appointments after the baseline visit.

**Table 1 jcm-10-03966-t001:** Baseline characteristics.

**No. of participants, n**	26
**No. of treated eyes, n**	34
**No. of binocular injections**	8
**No. of re-injections**	3
**Age when first injected, years (%)**	
<30	1 (3.8)
30–50	6 (23.1)
50–70	14 (53.8)
>70	5 (19.2)
Mean, years (SD)	58 (15.5)
**Gender, n (%)**	
Female	18 (69.2)
Male	8 (30.8)
**Study period post-FAc implantation**	
Mean, weeks (range)	18 (1–60)
**Uveitis classification, n (%)**	
Uveitis anterior with ME	1 (2.9)
Uveitis intermedia with ME	24 (70.6)
Uveitis posterior with ME	6 (17.6)
Panuveitis with ME	3 (8.8)
**Aetiology of non-infectious uveitis (eyes), n (%)**	
Idiopathic	17 (50.0)
Sarcoidosis	3 (8.8)
Multiple sclerosis	5 (14.7)
Rheumatoid arthritis	2 (5.9)
Acute zonal outer occult retinopathy (AZOOR)	1 (2.9)
Multifocal chorioretinitis and panuveitis (MCP)	2 (5.9)
Birdshot chorioretinopathy	2 (5.9)
Ocular tuberculosis (non-infectious when treated)	2 (5.9)
**Lens status, n (%)**	
Phakic	4 (11.8)
Pseudophakic	30 (88.2)
**Duration of macular edema prior FAc implantation, n**	
<3 years	7 (20.6)
>3–5 years	5 (14.7)
>5–10 years	13 (38.2)
>10 years	9 (26.5)
Mean, years (SD)	6.9 (4.0)
**Severity of macular edema, n (%)**	
CST < 300 µm	10 (29.4)
CST ≥ 300 µm	24 (70.6)
**Patients with local therapy for ME, n (%)**	
Intravitreal triamcinolone	5 (19.2)
Orbital floors steroids	24 (92.3)
Intravitreal dexamethasone	26 (100)
Surgically implanted fluocinolone acetonide 0.59 mg (off-label)	1 (3.8)
**Patients with systemic therapy, n (%)**	
None	12 (46.2)
Corticosteroid (CS) only	4 (15.4)
CS and immunomodulatory therapy	10 (38.5)

**Table 2 jcm-10-03966-t002:** Injections pre- and postinjection of the FAc implant.

	Eyes (n)		Injections (n)		Mean (n)		SD	
	pre FAc	post FAc	pre FAc	post FAc	pre FAc	post FAc	pre FAc	post FAc
Dexamethasone intravitreal	33	4	232	9	7	2.3	4.8	1.9
Triamcinolone periocular	30	3	84	6	2.8	2	1.7	1.7
Triamcinolone intravitreal	8	0	28	0	3.5		3.1	
Triamcinolone subconjunctival	8	3	19	10	2.4	3.3	1.2	2.3

**Table 3 jcm-10-03966-t003:** Systemic treatment pre- and post-FAc implantation.

	Pre FAc		Post FAc	
Systemic Treatment	n	Mean Dose	n	Mean Dose
Azathioprine	1	50 mg/d	1	50 m/d
Rituximab	1	1000 mg/a	1	1000 mg/a
Adalimumab	1	40 mg/2w	0	0
Ciclosporin	3	113 mg/d	2	100 mg/d
Mycophenolate	3	1080 mg/d	2	360 mg/d
Methotrexate	4	19 mg/w	1	10 mg/w
Prednisolone	6	13 mg/d	6	5 mg/d

**Table 4 jcm-10-03966-t004:** IOP change following FAc implantation.

IOP Change after FAc Implantation, n (%)	
IOP increase > 5 mmHg	7 (20.6)
IOP increase > 10 mmHg	2 (5.9)
IOP increase > 20 mmHg	0 (0)
Mean IOP increase, mmHg (SD)	4.4 (3.7)
**IOP lowering eyedrops**	
Number of eyes (n)	3 (8.8)
Number of eye drops, total	5
**IOP lowering surgery** Number of eyes (n)	0 (0)

## Data Availability

All available data generated or analyzed during this study are included in this published article.

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
