# Peer review of "Intravitreal 0.19 mg Fluocinolone Acetonide Implant in Non-Infectious Uveitis"

_jcm, 2021, doi:10.3390/jcm10173966_

Round 1

Reviewer 1 Report

Hikal et al. Intravitreal 0.19 mg fluocinolone acetonide implant in non-infectious uveitis. JCM. jcm-1341910. 100821 

Line 49ff: ‘However, there is currently a lack of real-world data on the use of the FAc implant in non-infectious uveitis affecting the posterior segment of the eye and so our retrospective observational study was designed to evaluate its effectiveness and safety in our practice.’

The author’s premises appears to be as stated here.

Line 54-55: ‘a retrospective chart review involving 34 eyes (26 patients) with non- infectious uveitic macular edema (UME) who were treated in the Department of Ophthalmology at the University of Heidelberg, Germany.’

The readership deserves to know which tests were carried out that excluded a known infective cause for the uveitis

Table 1, Line 82, does not tell the readership what other tests were done to render these cases being diagnosed as non-infectious.

Methods – The authors should tell the readership how the patients were selected for injections as opposed to simply topical therapy, orbital injections or subconjunctival injections.  

Table 2, Line 84ff: ‘subcunjunctival’ should read ‘subconjunctival’.

Line 75, 76, 94, 102, 103, 146, 182, 188 etc: It is not clear what these comments mean nor who made them. I presume they refer to various figures and tables.

Line 92: ‘macular’ should read ‘macula’.

Line 106: ‘triamcinolne’ should read ‘triamcinolone’.

Line 209: ‘an additional IOP drop was required’. This drop should be mentioned by name and the patient’s other drops documented.

Line 217: ‘foramen’. This should read ‘retinal tear’ or ‘retinal hole’.

Line 245: ‘the Huron study’: It is not clear to which article the authors are referring as this article does not appear to be in the References.

Line 323-4: Therapy with FAc implant enabled the injection intervals to be stretched, so that 2 eyes were injected with dexamethasone each at an interval of 6 and 12 months.

It should be asked what was the extend of cross-over was with the authors’ previous study on 11 eyes*. In other words, some of the patients may need to be cited.

(*Weber, L.F.; Marx, S.; Auffarth, G.U.; Scheuerle, A.F.; Tandogan, T.; Mayer, C.; Khoramnia, R. Injectable 0.19-mg fluocinolone acetonide intravitreal implant for the treatment of non-infectious uveitic macular edema. J. Ophthalmic Inflamm. Infect. 2019, 9, 3, doi:10.1186/s12348-019-0168-9.)

The authors should also state that the ultimate resolution of treating the patients was macular oedema. Therefore, the authors should be asked if this study adds anything new to the literature. If it does add something new, this should be eminently documentable.    

Author Response

Response to Reviewer 1 Comments

Point 1: Hikal et al. Intravitreal 0.19 mg fluocinolone acetonide implant in non-infectious uveitis. JCM. jcm-1341910. 100821

Line 49ff: ‘However, there is currently a lack of real-world data on the use of the FAc implant in non-infectious uveitis affecting the posterior segment of the eye and so our retrospective observational study was designed to evaluate its effectiveness and safety in our practice.’

The author’s premises appears to be as stated here.

Response 1: The text is changed to: "Our retrospective observations study was designed to evaluate the effectiveness and safety of the use of the FAc implant in non-infectious uveitis affecting the posterior segment of the eye in our practice in a rather long follow-up in a real-world setting."

Point 2: Line 54-55: ‘a retrospective chart review involving 34 eyes (26 patients) with non- infectious uveitic macular edema (UME) who were treated in the Department of Ophthalmology at the University of Heidelberg, Germany.’

The readership deserves to know which tests were carried out that excluded a known infective cause for the uveitis

Response 2: To exclude infectious uveitis, patients are tested for toxoplasmosis, syphilis, tuberculosis, herpes, cytomegalovirus, Lyme disease, or HIV, depending on the morphology of the uveitis. The information was added now to the manuscript.

Point 3: Table 1, Line 82, does not tell the readership what other tests were done to render these cases being diagnosed as non-infectious.

Response 3:  We now added in the Methods how we distinguish infectious from non-infectious uveitis.

Point 4: Methods – The authors should tell the readership how the patients were selected for injections as opposed to simply topical therapy, orbital injections or subconjunctival injections. 

Response 4: Patients with non-infectious uveitic macular edema get intravitreal FAc implantations reimbursed and the drug has a label for this disease. Thus, the FAc implantation was therefore preferred to simple topical therapy, orbital injections, or subconjunctival injections.

Point 5: Table 2, Line 84ff: ‘subcunjunctival’ should read ‘subconjunctival’.

Response 5: This mistake has been corrected.

Point 6: Line 75, 76, 94, 102, 103, 146, 182, 188 etc: It is not clear what these comments mean nor who made them. I presume they refer to various figures and tables.

Response 6: There was a recurrent error in the references of the figures, due to the editing software, which is fixed now.

Point 7: Line 92: ‘macular’ should read ‘macula’

Response 7: This mistake has been corrected.

Point 8: Line 106: ‘triamcinolne’ should read ‘triamcinolone’.

Response 8: This mistake has been corrected.

Point 9: Line 209: ‘an additional IOP drop was required’. This drop should be mentioned by name and the patient’s other drops documented.

Response 9: The first patient was initially treated with brinzolamide. The therapy was subsequently supplemented with brimonidine and then latanoprost. The second patient had no eye drops at the time of injection. After pressure elevation in the right eye, dorzolamide was subsequently required. The information was added to the manuscript.

Point 10: Line 217: ‘foramen’. This should read ‘retinal tear’ or ‘retinal hole’.

Response 10: This mistake has been corrected.

Point 11: Line 245: ‘the Huron study’: It is not clear to which article the authors are referring as this article does not appear to be in the References.

Response 11: The Huron study by Lowder, C.; Belfort, R.; Lightman, S.; Foster, C.S.; Robinson, M.R.; Schiffman, R.M.; Li, X.-Y.; Cui, H.; Whitcup, S.M. Dexamethasone intravitreal implant for noninfectious intermediate or posterior uveitis. Arch. Ophthalmol. 2011, 129, 545–553, doi:10.1001/archophthalmol.2010.339 which is described in more details in the following lines, is as in the text under point 5 in the references.

Point 12: Therapy with FAc implant enabled the injection intervals to be stretched, so that 2 eyes were injected with dexamethasone each at an interval of 6 and 12 months.

It should be asked what was the extend of cross-over was with the authors’ previous study on 11 eyes*. In other words, some of the patients may need to be cited.

(*Weber, L.F.; Marx, S.; Auffarth, G.U.; Scheuerle, A.F.; Tandogan, T.; Mayer, C.; Khoramnia, R. Injectable 0.19-mg fluocinolone acetonide intravitreal implant for the treatment of non-infectious uveitic macular edema. J. Ophthalmic Inflamm. Infect. 2019, 9, 3, doi:10.1186/s12348-019-0168-9.)

The authors should also state that the ultimate resolution of treating the patients was macular oedema. Therefore, the authors should be asked if this study adds anything new to the literature. If it does add something new, this should be eminently documentable.   

Response 11: The 11 eyes from Weber et al. were included in this work. These were the first eyes injected with FAc off-label at that time. These 11 eyes are also the ones with the longest follow-up periods. During this time, we were able to collect more data on each case and include them in the current work. Case No. 6 of Weber et al. is the patient with the longest follow-up time. Cases 3 and 4 required re-injection. Now, we mentioned in the text in section 3.1.2. that 2 of these re-injections were already evaluated in the work of Weber et al.

It is now mentioned in the manuscript that a rather long follow-up of patients treated with an FAc implant in a real-world setting is presented.

Reviewer 2 Report

The authors present interesting real world data validating the efficacy of intravitreal 0.19 mg fluocinolone implants for non-infectious uveitis.

  • Table 2 - when presenting the pre- and post-FAc treatments, were these results annualized?  There is a lot more date pre-FAc (up to 13 years) then post (up to 7 years) that could skew the results in favor of post-FAc.
  • Figure 1 - when was the complete drying or reduced CRT noted in relation to the FAc?  Was it a standard interval or at any given time?  Could the authors clarify what proportion of this drying effect was seen with FAc monotherapy as opposed to additional therapy in conjunction with FAc?  What proportion of eyes had a history of complete drying in the past with other agents compared to drying after FAc?  Figure 2 gives a better sense of the OCT response over time - maybe the authors would consider presenting Figure 1 in a similar manner.
  • Figure 5 - similarly to Figure 1, this figure does not give a sense of the time element, critical in a long-acting agent. Is the maximum change in vision achieved early, mid-course, or late? Do eyes that gain 3+ lines of vision maintain this gain or drift back to baseline?  How much of these visual acuity changes can be attributed purely to FAc versus additional therapy?
  • Figure 6 - when was post FAc activity first assessed, when was the maximum benefit attained, and how long was it maintained prior to additional therapy?

Author Response

Point 1: Table 2 - when presenting the pre- and post-FAc treatments, were these results annualized?  There is a lot more date pre-FAc (up to 13 years) then post (up to 7 years) that could skew the results in favor of post-FAc.

Response 1: It is now mentioned in the discussion as a limitation of the study, that there is more pre-FAc data (up to 13 years) then post-FAc (up to 7 years) that could skew the results in favor of post-FAc.

Point 2: Figure 1 - when was the complete drying or reduced CRT noted in relation to the FAc?  Was it a standard interval or at any given time?  Could the authors clarify what proportion of this drying effect was seen with FAc monotherapy as opposed to additional therapy in conjunction with FAc?  What proportion of eyes had a history of complete drying in the past with other agents compared to drying after FAc?  Figure 2 gives a better sense of the OCT response over time - maybe the authors would consider presenting Figure 1 in a similar manner.

Response 2: The data mentioned in Figure 1 refer to the effect on macular edema after 1-3 months by single injection of FAc without additional therapy divided intothree groups (dry, less with residual fluid and no improvement). Figure 2 was intended to show the course of CRT. The missing information for Figure 1 was only mentioned in the text not in the description of the figure. This has now been corrected.

Regarding macular edema prior to injection with FAc: It is difficult to make a general statement because the therapy regimen was highly individualized. Reduction of macular edema and dryness was achieved by injections, but the intervals between injections were shortened and prolonged in an individualized way, so that a general statement would be very  difficult to make.

Point 3: Figure 5 - similarly to Figure 1, this figure does not give a sense of the time element, critical in a long-acting agent. Is the maximum change in vision achieved early, mid-course, or late? Do eyes that gain 3+ lines of vision maintain this gain or drift back to baseline?  How much of these visual acuity changes can be attributed purely to FAc versus additional therapy?

Response 3:  We here present the maximum visual acuity gain in comparison to baseline. We completely agree that it is not possible to that these visual acuity changes can be attributed purely to FAc and/or additional therapy. We therefore added this as a limitation of the study.

Point 4: Figure 6 - when was post FAc activity first assessed, when was the maximum benefit attained, and how long was it maintained prior to additional therapy?

Response 4: The post FAc data mentioned in Figure 6 refer to the activity state that already occurred in the first examinations 1-3 months after injection.

Reviewer 3 Report

Thanks you for you work. It's very interesting and useful for the clinician.

The patients, methods and results are clearly described.  The results are useful for the clinician. I have no major criticism. There is a recurrent error in the references of the figures, probably due to the editing software. 

Author Response

Thank your very much for the positive feedback.

The reference error has been corrected.

Round 2

Reviewer 1 Report

Table 1: ‘Birdshot’ should read ‘Birdshot chorioretinopathy’.

Line 97: ‘were identified’ should read ‘was identified’ as ‘total’ is a collective noun.

Ich glaube dass dies der identische Ausdruck auf Deutsch sei.

Line 113: ‘injection and remains dry’ should read ‘injection, the macula remaining dry…’

Line 221-222: ‘neither the right nor the left eye was treated with IOP lowering eye drops’. The authors need to confirm that neither eye, of this patient who had been treated bilaterally, actually had a raised pressure that needed treating, or whatever actually really happened.

Author Response

Point 1: Table 1: ‘Birdshot’ should read ‘Birdshot chorioretinopathy’.

Response 1: Birdshot chorioretinopathy has been corrected in the table.

Point 2: Line 97: ‘were identified’ should read ‘was identified’ as ‘total’ is a collective noun.

Response 2: The mistake has been corrected.

Point 3: Line 113: ‘injection and remains dry’ should read ‘injection, the macula remaining dry…’

Response 3: It has been corrected in the text.

Point 4: Line 221-222: ‘neither the right nor the left eye was treated with IOP lowering eye drops’. The authors need to confirm that neither eye, of this patient who had been treated bilaterally, actually had a raised pressure that needed treating, or whatever actually really happened.

Response 4: It has been added to the text that the IOP rise in the left eye did not require additional Treatment.

Reviewer 2 Report

Minor comments/suggestions:

Lines 92/93 -

Table 2 is shown twice

Line 128 -

Consider replacing "further" with "additional" to avoid its use twice in the same sentence: Weber et al. mentioned this patient as case 4, whose additional data were further processed in this work [12].

Line 382 - 

Consider clarifying: This might skew the results in favour of post-FAc in terms of mean number of treatments pre and post Facebook injection.

Author Response

Point 1: Lines 92/93 - Table 2 is shown twice

Response 1: The table was changed and therefore the first version was deleted from the manuscript. Due to tracking, the old table can still be seen.

Point 2: Line 128 - Consider replacing "further" with "additional" to avoid its use twice in the same sentence: Weber et al. mentioned this patient as case 4, whose additional data were further processed in this work [12].

Response 2: The change has been made as suggested.

Point 3: Line 382 - Consider clarifying: This might skew the results in favour of post-FAc in terms of mean number of treatments pre and post FAc injection.

Response 3: The change has been made as suggested.